# Otolith "spawning zones" across multiple Atlantic cod populations: Do they accurately record maturity and spawning?

**Côme Denechaud**[1,2]*, **Audrey J. Geffen**[1,2], **Szymon Smoliński**[1,3], **Jane A. Godiksen**[1]

**1** Demersal Fish Research Group, Institute of Marine Research (HI), Bergen, Norway, **2** Department of Biological Sciences, University of Bergen (UiB), Bergen, Norway, **3** Department of Fisheries Resources, National Marine Fisheries Research Institute, Gdynia, Poland

* come.denechaud@hi.no

**Data Availability Statement:** The datasets generated and analyzed during the current study

## Abstract

Specific changes identified in the otolith macrostructure of Northeast Arctic cod as "spawning zones" are presumed to represent spawning events, but recent experimental studies have challenged this relationship. Because these zones are not routinely recorded outside of Norway, otoliths from multiple Atlantic cod populations with different life history and environmental traits were first examined to see if spawning zones could be identified as a general characteristic of cod. Then, a large archival collection of cod otoliths was used to investigate temporal changes in the occurrence of spawning zones and test for correlations between maturity at age derived from otolith spawning zones and gonad maturity stages. This study shows that spawning zones likely are a universal trait of Atlantic cod and not limited to certain environments or migratory behaviors as previously proposed. Maturity at age derived from spawning zone data showed trends consistent with those from gonad examinations. However, spawning zones appear to form with a one- or two-year lag with sexual maturity, which is suspected to reflect a stabilizing of energy partitioning after the first spawning events. Our results illustrate the potential for use of spawning zones, for example in species or populations with limited available maturity data, and highlights the need for addressing the physiological processes behind their formation.

## Introduction

Otoliths have been widely used in fisheries research since the mid-20<sup>th</sup> century [1]. Otolith growth increments result from rhythmic variations in the accretion rates of both calcium carbonate crystals and an organic matrix [2], which is influenced by variability in temperature, food consumption and metabolism. Characteristic macroscopic features identified as optical differences between opaque (more organic) and translucent (less organic) zones are in turn traditionally associated with periods of high and low growth [3]. In many fish species, these increments may correspond to growth patterns linked to seasonality over a single year and consequently reflect individual responses over their lifetime [4]. Counting of increments can

are available at the Norwegian Marine Data Center repository (https://doi.org/10.21335/NMDC-480969829).

**Funding:** Funding for this work was provided by the Icelandic Research Fund Grant 173906-051. The funders had no role in study design, data collection and analysis, decision to publish, or preparation of the manuscript.

**Competing interests:** The authors have declared that no competing interests exist.

provide estimates of individual age and thus otolith reading is central to fisheries science, where the data produced is often used to infer demographic structure primarily for management and conservation [5].

Since biomineralization processes are influenced by both environmental and physiological factors, otoliths may also act as recorders of life history events that can interrupt somatic growth. These features in the otolith zonation are traditionally found in otolith microstructures and are often referred to as "checks" [6]. However, checks in the otolith zonation also appear at the macrostructural level in response to significant life history events, irrespective of fish age. These checks have been described in several species for example in relation to migration or stress [7], but more importantly to maturity [8] and reproduction [9]. In many exploited populations, individual maturity is generally estimated using gonad maturity stages read during scientific surveys or on-board fishing vessels. Because survey effort and gonad interpretation are not necessarily consistent through time and between populations or institutes, the property of otoliths to record life history events such as maturity and spawning could therefore be helpful for data-poor populations or for reconstructing historical time series when otolith archives are available.

The annual reproductive cycle in Atlantic cod (*Gadus morhua*) usually starts with gonad development during the autumn and spawning takes place in early spring of the following year [10, 11]. However, there are many distinct populations or stocks of cod inhabiting different habitats across the north Atlantic, and they vary considerably in life history characteristics. For example, cod from the North Sea experience warmer conditions with significant seasonal variations associated with a fast life history where maturation occurs as early as age 3 and lifespan seldom exceeds 10 years [12]. On the other hand, cod from the Barents Sea experience colder conditions with reduced temperature range and seasonality and consequently shows a much slower life history, where fish only become mature at age 7 to 8 and may live several decades [13].

Rollefsen [14] first identified specific changes in the otolith macrostructure of Northeast Arctic (NEA) and Norwegian coastal cod (NCC) populations inhabiting the Barents Sea and the Norwegian coast, which he subsequently referred to as "spawning zones". He described a sudden alteration of the otolith annual zonation pattern where the first 6–10 inner increments are followed by increments which are much narrower and regular in width, with a proportionally thicker translucent zones relative to opaque zones in comparison to the inner increments. Rollefsen also proposed that these zones reflected years following sexual maturity where spawning took place, and they have since been regularly recorded during NEA and NCC cod data collection for assessment. Spawning zone data contribute to the investigation of maturity and population age-structure of NEA cod and NCC [15]. However, the mechanisms of so-called "spawning zone" formation have not been determined, and their validity as a marker of sexual maturity is now under debate. It was initially assumed that the narrowness of spawning zones might reflect the trade-offs between growth and reproduction [16], but recent experimental studies have shown that physiological changes associated with maturation and spawning alone are not sufficient to consistently induce spawning zones in cod otoliths [17]. Other factors associated with NEA cod spawning, such as the long-distance migration [18, 19] and/or changes in food intake and energy use [20, 21] may therefore also be the proximate cause for the formation of these distinct zones.

The assumption that these spawning zones reflect individual spawning events is further challenged by the high incidence of skipped spawning (up to 40%) in (female) cod [11, 22]. Indeed, many fish species, including Atlantic cod, are known to skip spawning in years when energy reserves are limited [23]. Alternatively, fish may skip spawning to promote further reproductive output by reinvesting energy into somatic growth and increased fecundity [24].

However, there is only anecdotal evidence of interruption in the sequence of spawning zones in cod otoliths. In comparison, skipped spawning events are recorded in the scale annuli of anadromous Atlantic salmon [25]. After attaining maturity at sea, Atlantic salmon undergo a demanding upstream migration during which the fish do not feed, causing partial resorption of the scales and the appearance of a specific check which is absent when the fish skips spawning and spends an extra winter at sea. Irgens *et al.* [17] suggested that the extended rest periods and low feeding/energy levels generally associated with skipped spawning in cod could result in the formation of "normal" spawning zones and explain the uninterrupted pattern observed. Indeed, the discrepancy between observed "spawning zones" and their chemical composition, which more accurately tracks spawning migrations in some species [26], indicates that spawning zones are likely not an identifying character of actual reproduction events but at best of the number of years since the onset of maturity.

The spawning zones of cod otoliths are not widely recognized or used for stock assessment in other cod populations outside the Barents Sea. It is difficult to define exactly what they represent. If the origin of spawning zones is strictly physiological and tied to the onset of sexual maturity and spawning, they would likely be detectable in any individual mature Atlantic cod regardless of population. On the other hand, if behavioural or environmental changes are partly responsible and if spawning zones are population-specific, there is a need to determine what underlying mechanisms are responsible for their appearance and how they correlate with maturity and reproduction.

In this study, we first collected otolith images from multiple Atlantic cod populations with different life history and environmental traits to determine if specific otolith features similar to the NEA cod spawning zones could be consistently identified as a general characteristic of cod, irrespective of life history patterns. Then, we used data from a large archival collection of NEA cod otoliths collected since the 1930s to investigate temporal patterns in the occurrence of spawning zones. We tested for correlations between maturity at age derived from otolith spawning zones and gonad maturity stages. Reader consistency across time was evaluated to determine whether there have been changes in the characterization of spawning zones. This study takes the first standardized approach to the evaluation of spawning zones in relation to maturity across spatial/population and temporal scales.

## Materials and methods

### Population selection and collection of otolith images

Extensive archives of cod otoliths collected from research vessels and commercial fishing exist for multiple fish stocks, often going back many decades. In this study, we aimed to investigate whether spawning zones could be reliably identified in multiple Atlantic cod populations from different environmental conditions and life histories. We therefore selected six Atlantic cod populations (**Table 1**): Northeast Arctic cod (NEA), Norwegian coastal cod (NCC), Icelandic cod (ICE), Cape Breton cod (CAN), West Greenland cod (GRE) and North Sea cod (NS).

A large collection of NEA cod otoliths was retrieved, processed and imaged at the Institute of Marine Research (Norway), the full methodology is available in Denechaud *et al.* [27]. ICE cod otoliths were selected and processed as described in Smoliński *et al.* [28]. NCC otoliths from Porsangerfjord were selected from a collection of sections partly described in Andrade *et al.* [29] and were imaged similarly to the NEA cod otoliths. For the other populations, images of previously read sectioned otoliths were provided with the associated fish data by different institutes. NS samples consisted of a mixture of otolith images provided by the Centre for Environment, Fisheries and Aquaculture Science (CEFAS) and described in Høie *et al.* [30] for the southern North Sea region defined by the International Council for the

**Table 1. Sampling area and environmental and life history traits of the cod populations whose otoliths were used in this study.**

| Cod populations | Code | Sampling area | Temperature range (mean) | Median age at maturity | Spawning period | Migratory behavior | Years | N |
|---|---|---|---|---|---|---|---|---|
| North Sea [32] | NS | North Sea | 8–16˚C (**10.5˚C**) | 2.71 years | January—May | Limited | 2000–2008 | 205 |
| Cape Breton [31] | CAN | Southern Gulf of St Lawrence | -1.5–17˚C (**8˚C**) | 4.46 years | May—June | Inshore / offshore | 2000 | 203 |
| West Greenland [33] | GRE | Sisimiut and Godthaabsfjord | -1.5–3˚C (**1.5˚C**) | 4,99 years | February—April | Inshore / offshore | 1974–2010 | 240 |
| Iceland [34] | ICE | South west Iceland | 3–12˚C (**4.5˚C**)* | 6.56 years* | January—June | Long-distance | 1929–2015 | 250 |
| Norwegian coastal [35] | NCC | Porsangerfjord | 2–10˚C (**5˚C**) | 5.53 years | January—April | Limited | 1996–2013 | 103 |
| Northeast Arctic [35] | NEA | Lofoten | 2–6˚C (**4.5˚C**)* | 6.85 years* | January—April | Long-distance | 1932–2015 | 250 |

Sampling area indicates where fish samples were collected. Range of sea temperatures experienced throughout a year and median age at maturity refer to the period of collection. Reference citations in the first column refer to official stock assessment reports presenting environmental conditions and biology for each cod population. *Century-long collections of samples were available for Norway and Iceland. Because maturity and environment changed significantly throughout the last 90 years, age-at-maturity and temperatures provided refer to the period 2000–2015.

Exploration of the Sea (ICES) as area IVc, and otolith images from an European Fish Ageing Network (EFAN) international reading exchange for the central and northern regions (ICES areas IVb and IVa). CAN samples were provided from the otolith reference collection from the Canadian Science Advisory Secretariat [31]. GRE samples from Sisimiut and Godthaabsfjord were provided from the otolith collection of Greenland Institute of Natural Resources.

### Identification and definition of spawning zones

Spawning zones have been formally defined and characterized for NEA cod and NCC [14, 15] and the same visual criteria were applied to identify changes in the otolith zonation pattern that could correspond to a spawning zone for the other populations. Spawning zones were defined as increments that: 1) appeared abruptly narrower, often more regular, and extending without interruption to the edge of the otolith; 2) had proportionally wider translucent zones relative to opaque zones, making them appear darker and more contrasted under reflected light; and 3) whose visual differences could be reliably identified on all growth axes, i.e. distal, ventral and dorsal (**Fig 1**). These identification criteria were verified by comparing spawning and non-spawning zones formed at the same age on a subsample of otoliths from 8-year-old NEA cod whose spawning zones had been identified by trained age readers.

Experienced otolith age readers in Norway have routinely read and interpreted spawning zones in NEA and NCC cod. The investigation of spawning zones in North Sea, Icelandic, Greenland and Cape Breton cod has however never been reported before, and a reader precision test was therefore carried out. A randomized blind subsample of 41 otolith images taken across all 6 populations was read on three occasions by both the study reader and an external reader experienced with NEA cod and NCC spawning zones.

### Comparison of spawning zone occurrence and maturity across cod populations

Since spawning zones are assumed to be linked to the onset of sexual maturity and individual spawning events, we used otoliths from a range of age classes starting around the mean age of maturity (**Table 2**). For NEA and Icelandic cod where a larger collection of samples was

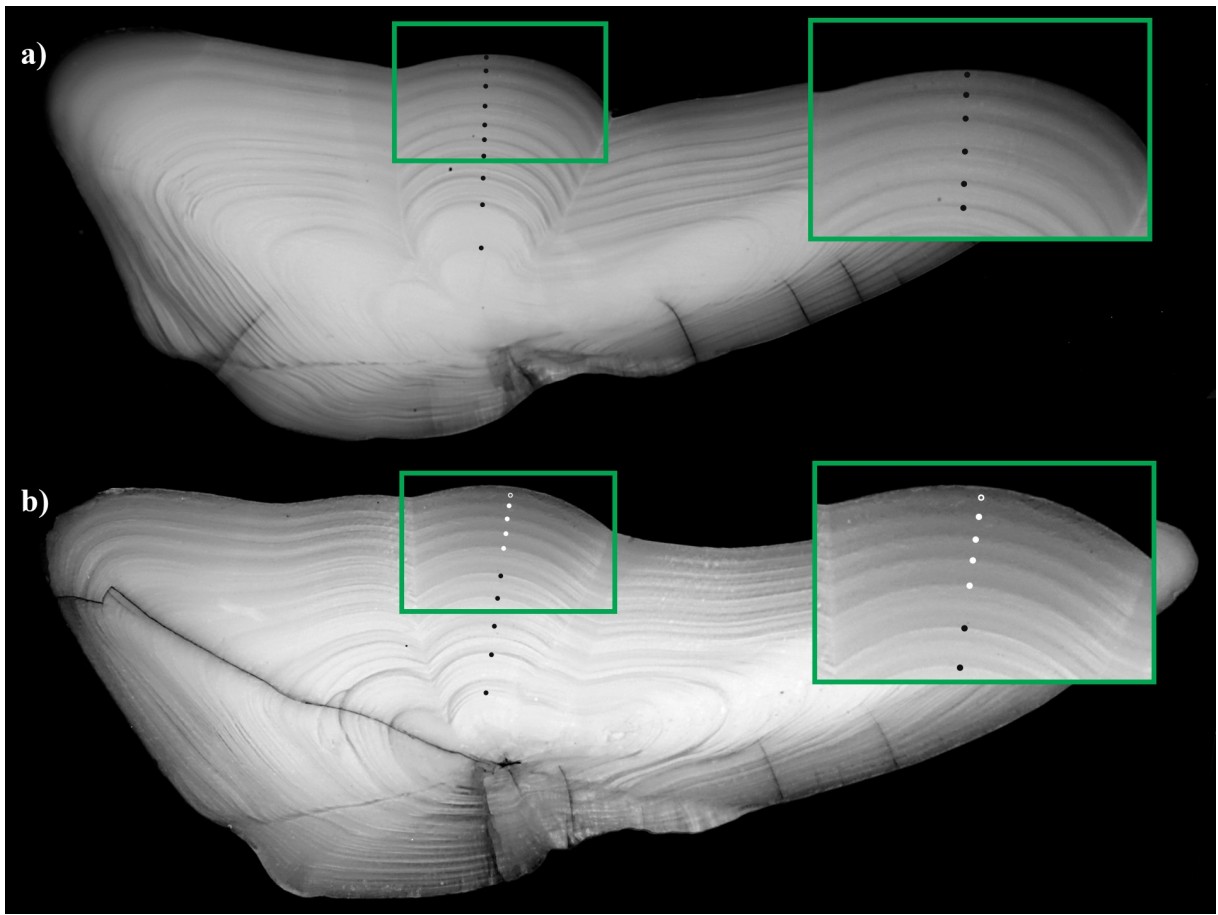

**Fig 1. Comparison of spawning zone absence and presence in two NEA cod otoliths.** a) 10-year-old fish with no visible spawning zones; b) 10-year-old fish with five visible spawning zones, identified as a cluster of similar-looking narrow increments with significantly wider translucent zones. Black circles mark regular growth increments; white circles mark spawning zones as described above; semi-filled white circles indicate an incomplete marginal increment that would likely be a spawning zone when fully formed.

**Table 2. Age distribution of the otolith images read from each population.**

|  | Population | | | | | |
|---|---|---|---|---|---|---|
| Age | NS | CAN | GRE | ICE | NCC | NEA |
| 4 | 149 |  |  |  |  |  |
| 5 | 11 | 2 |  |  | 2 |  |
| 6 | 26 | 51 | 151 |  | 35 |  |
| 7 | 12 | 46 | 54 | 9 | 33 | 14 |
| 8 | 5 | 71 | 22 | 157 | 21 | 181 |
| 9 | 1 | 26 | 10 | 50 | 7 | 12 |
| 10 | 1 | 6 | 3 | 19 | 3 | 14 |
| 11 |  | 1 | 1 | 7 | 2 | 20 |
| 12 |  |  |  | 8 |  | 9 |
| TOTAL | 205 | 203 | 240 | 250* | 103 | 250* |

Detailed numbers of samples are given per population and age class.

*Subsample randomly selected between 1986 and 2015 from the entire collection of samples available.

available, a random subsample of 250 otoliths collected between 1986 and 2015 was selected, to limit the potential effects of temporal changes in age at maturity on the presence and number of spawning zones. A total of 1251 otoliths images were read and measured for comparison.

The sample processing and imaging methodology differed between populations and the images therefore varied in terms of resolution, mode (greyscale or color) and orientation. To ensure that otolith zonation was interpreted consistently for all populations, only images taken under reflected light were used in the analysis and minor visual corrections were applied when necessary. Every image was read and annual increments measured by a single reader using a dedicated set of ObjectJ macros, a plugin for ImageJ [36], following the workflow detailed in Denechaud *et al*. [27].

We assessed the proportions of mature fish for each age class and population based on the presence or absence of spawning zones in our samples, and compared it to the mean proportions of mature fish at each age calculated from the relevant stock assessment reports (**Table 1** & **S1 Fig**) [31–35]. Because each spawning zone increment reflects an individual spawning in that year, a given fish will contribute as many times and years to the spawning stock biomass as the number of spawning zones read in its otoliths. Therefore a separate set of maturity ogives was generated based on individual spawning zones to take into account the multiple annual contributions from each individual to the mature portion of the population. This produced a distribution of 9025 individual increments identified by year and by age class from the original 1251 fish (**Table 2**). Each increment was then assigned a maturity status (immature or mature) based on whether that increment was identified as a spawning zone or not. Then, for each age class and population, the proportions of individual increments identified as spawning zones were expressed as the proportion of fish that were mature at that age and compared to the maturity ogives extracted from the literature.

## Long-term analysis of maturity and spawning zone occurrence in NEA cod

Temporal variations in the spawning zone record, and its relation to direct assessment of maturity based on examination of gonad maturity stages [37], was investigated using data from a century-long collection of NEA cod otoliths. Atlantic cod otoliths from both research vessels and commercial fishing since the late 1920s are archived together with data on general location, date, fishing gear and biological information (length, weight and sex). In particular, the information about age at first maturity and alleged spawning events interpreted from spawning zones has been recorded for most individuals since 1932. We extracted the recorded biological information from the archive database on 382 218 NEA cod and their otoliths, collected in the Barents Sea and Lofoten between 1932 and 2015, and used it as basis for our reconstruction of population maturity trends over time.

Spawning zones had already been previously used to reconstruct NEA cod maturity ogives for the period 1932–1982, before regular surveys and use of direct examination of gonad maturity stages became the basis of standardized maturity assessment [38, 39]. Gulland [40] first developed a method to reconstruct maturity ogives using only age-specific proportions of first-time spawning individuals derived from otolith spawning zones readings, which was then expanded on by Jørgensen [41]. Since 1982 however, maturity ogives are fully derived from gonad maturity stage data assessed with data from the combined Barents Sea winter surveys and from the Lofoten survey [13, 39]. Here, we instead applied the approach previously explored in our population comparison to assign individual otolith increments from each fish from the archive database to their calendar year of formation, in order to represent their contribution at each age of their life. This allowed us to utilize information on the yearly

contribution of individual fish to the mature part of the stock throughout their life. The 382 218 fish from the otolith archive were transformed into a total of roughly 2 946 000 individual increments spanning 1910 to 2015. Each increment was then assigned a maturity status (immature or mature) based on its status as a spawning zone, as determined by age readers at the time. Then, for each calendar year and age class, the proportions of individual increments identified as spawning zones were calculated and compared to the proportions of mature at age extracted from stock assessment literature.

We then used the calculated proportions of mature fish at age for each year to generate maturity ogives and explore changes in maturity thresholds over time. Because maturity data is binomial in nature (either a fish is mature or is not), maturity ogives are traditionally represented as sigmoid curves that conform to a logistic regression, which can be expressed as:

$$\log\left(\frac{\rho}{1-p}\right) = \alpha + \beta_1 X$$

where $p$ is the probability of fish being mature, $X$ is the explanatory variable of interest (here age, but usually also length), and $\alpha$ and $\beta_1$ are the coefficients of the regression. From this equation, we used a binomial Generalized Linear Model (GLM) with logit link function to model the maturity ogives as a function of age for each year, and calculated specific maturity thresholds for comparison: A25, A50, A75 (respectively the ages at which 25/50/75% of the fish are mature).

Finally, a subset of otoliths was examined to check for variation in reader interpretation over time. A large selection of otoliths collected between 1932 and 2015, that had been previously imaged and analysed by Denechaud *et al.* [27]), were reviewed by the same reader and the number of spawning zones for all these samples evaluated. Because the subset was mainly composed of 8 year old fish, this age class was selected for the comparison as it was present in every year and was consistently represented at a relatively high proportion in the archive database, despite the large reduction in older age classes observed since the 1950s. A total of 2979 age 8 otoliths were re-evaluated. We compared new and old spawning zone readings to quantify how the interpretation of spawning zones varied through time between readers from different periods where the biological parameters of the population were different.

## Results

### Definition and comparison of spawning zone occurrence across cod populations

The visual criteria commonly used for determining spawning zones (narrower with a larger translucent relative to opaque zone) were checked on a random subsample of 56 previously read NEA cod otoliths. Increments assigned as spawning zones were significantly narrower (t (26) = 2.18, n = 55, p = 0.04) and had a proportionally wider translucent zone relative to opaque zone (t(40) = -10.74, n = 55, p<0.001) than non-spawning zones formed at the same age (**Fig 2**).

Spawning zones could be reliably identified by visual inspection of the otoliths in all six populations of Atlantic cod (**Fig 3**). Comparisons of two readers on a randomized blind subsample confirmed that the interpretation of spawning zones was generally consistent between readings of the same reader (**S1 Table**). Internal consistency between all three readings was relatively high in both reader at respectively 90.2% (CV = 7.3%, SD = 0.06) and 75.6% (CV = 22.1%, SD = 0.18) agreement. When comparing median number of spawning zones across all three readings, both readers had a relatively high agreement (70.7%, CV = 24.1%, SD = 0.33) and successfully identified spawning zones in all cod populations, although with lower agreement in some.

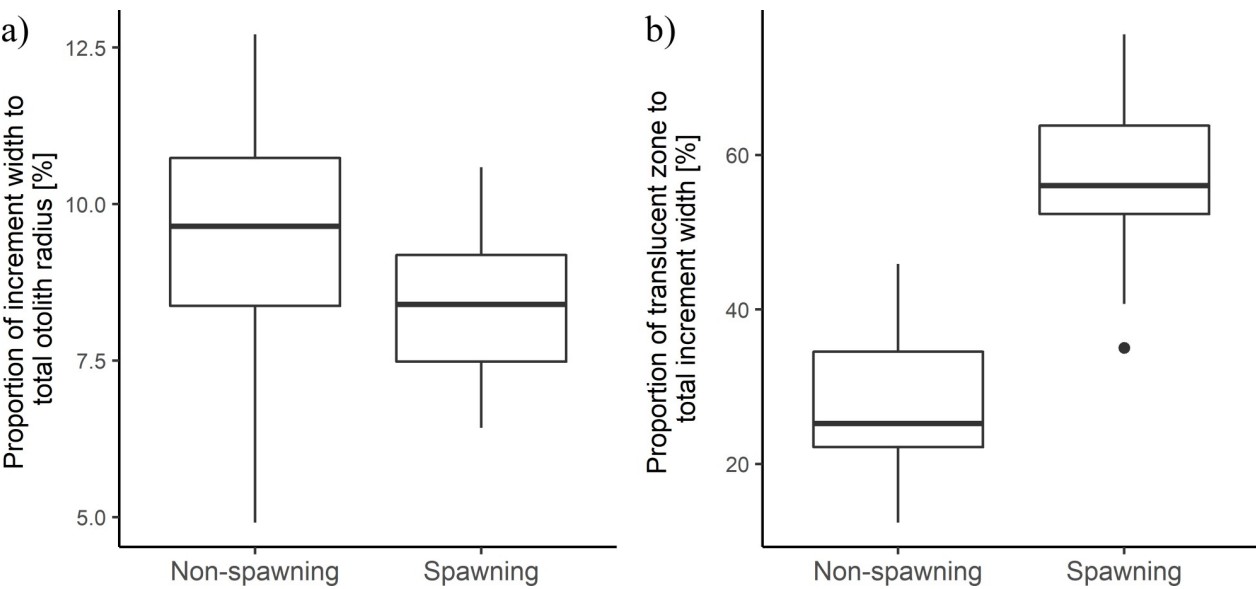

**Fig 2. Comparison of width of increments formed at the same age and identified as either non-spawning or spawning zones.** An age-verified subsample of 8-years old NEA cod otoliths was used for the comparison.

Furthermore, the number of spawning zones increased with fish age in all populations and higher proportions of older fish had multiple spawning zones (Fig 4). There was a general concordance between the maturity ogives and the age at maturity indicated by spawning zones

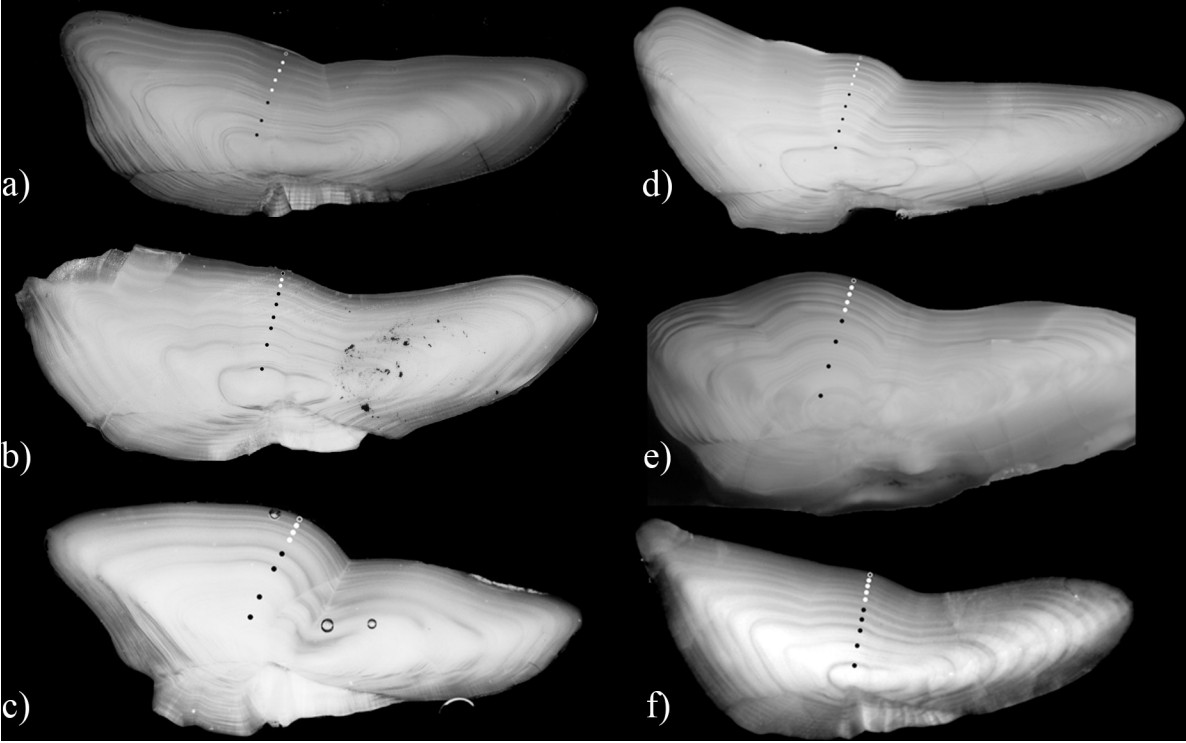

**Fig 3. Annotated examples of otoliths with visible spawning zones.** NEA cod (a), NCC (b), NS cod (c), Icelandic cod (d), Greenland cod (e) and Cape Breton cod (f). Spawning zones shown as white circles.

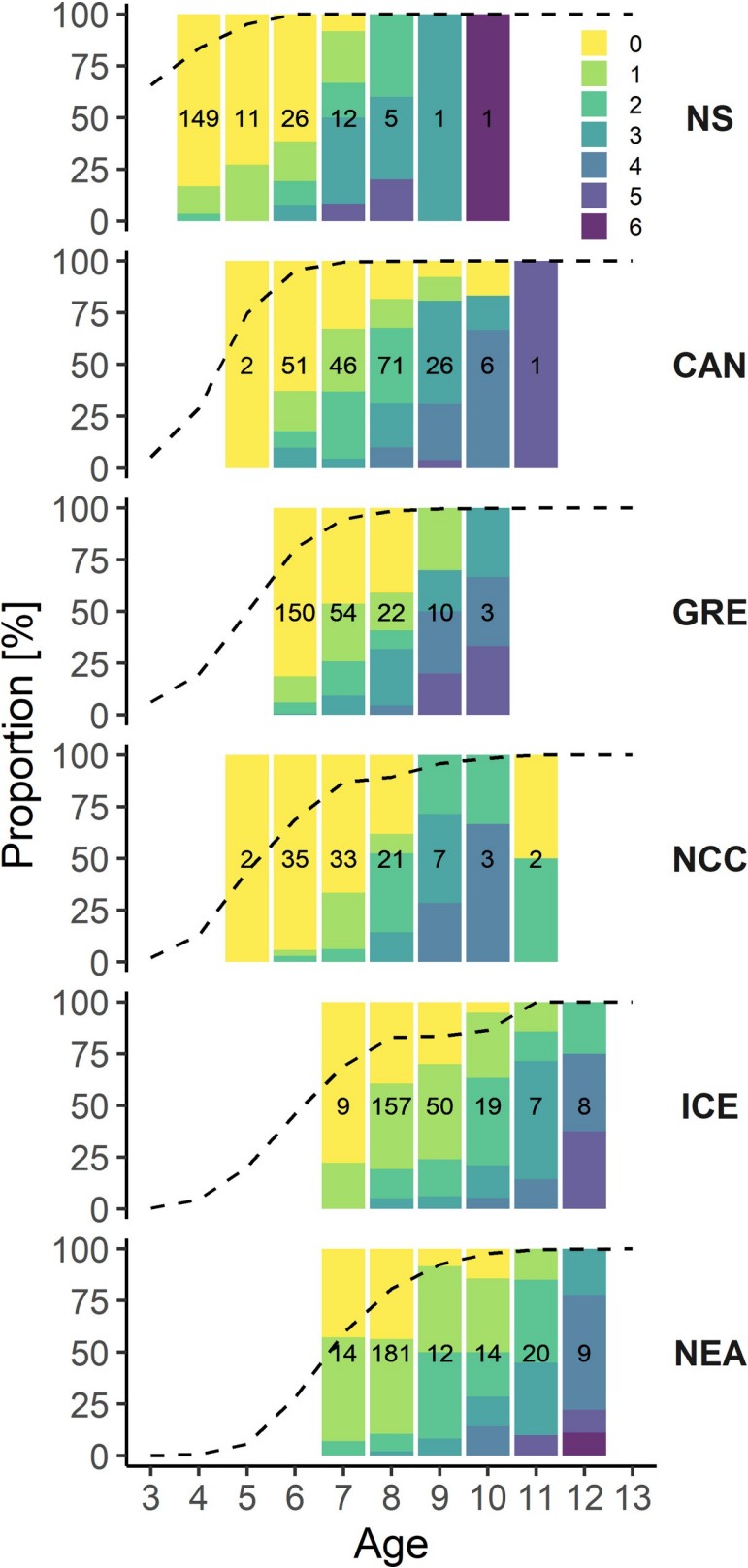

**Fig 4. Proportions of otoliths and spawning zones identified per age class and population.** Color scale refers to the number of spawning zones counted. Column numbers indicate the number of otoliths available from each age class for

each population. Dashed lines represent the proportions of mature at age calculated from the maturity ogives from stock assessment data for each population in the period of sample collection.

occurrence across all populations (**Fig 4**). For example in NS cod, which matures at age 3–4, all 8-year old fish had at least one spawning zone. On the other hand in NEA cod, which matures at age 6–7, 45% of 8-year old fish had no spawning zones and only 10.5% had more than one.

For the older age classes, where the proportion of mature fish was close to 1, the proportion of fish with at least one spawning zone aligned relatively well with the mean proportions of maturity at age calculated in the stock assesment ogives (**Fig 5**). However, with the exception of NEA cod where the two proportions remained relatively similar, spawning zones consistently showed a lower proportion of mature fish for most of the intermediate age classes. The relationship between both maturity proportions showed a distinct convex shape, particularly pronounced in the earlier-maturing populations such as NS, CAN and GRE, where age classes at both extremes (either <10% or >90% of fish mature) had a very high agreement between spawning zone and assessment data, but proportion of mature fish in the intermediate age classes was consistently underestimated using spawing zones (**Fig 5**). The agreement was much better in Icelandic and NEA cod.

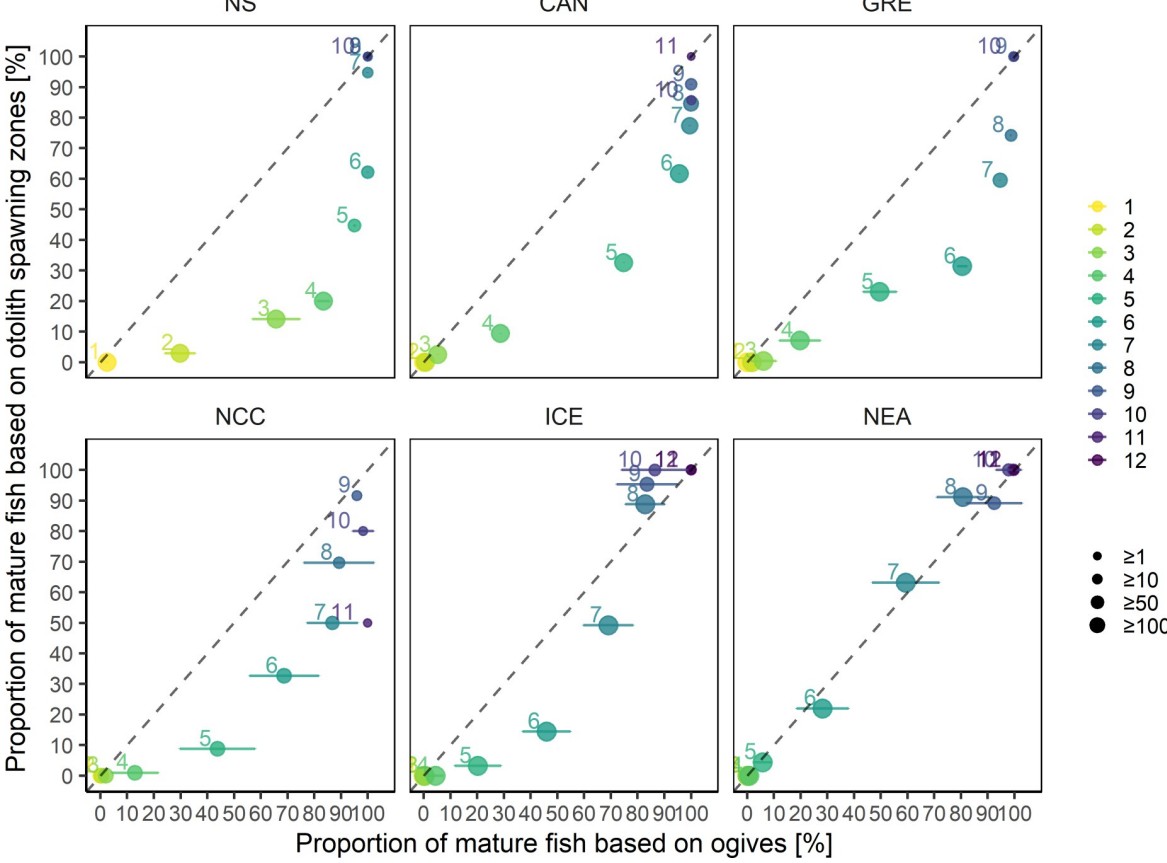

**Fig 5. Proportions of mature at age (colors) calculated for each population from the visual identification of individual spawning zones, compared to the mean proportions of mature at age calculated from the maturity ogives for the period specific to each population.** Horizontal bars indicate the standard deviation of the mean of each proportion mature at age over the range of years defining the samples. Point size is proportionate to the number of individual increments per age class.

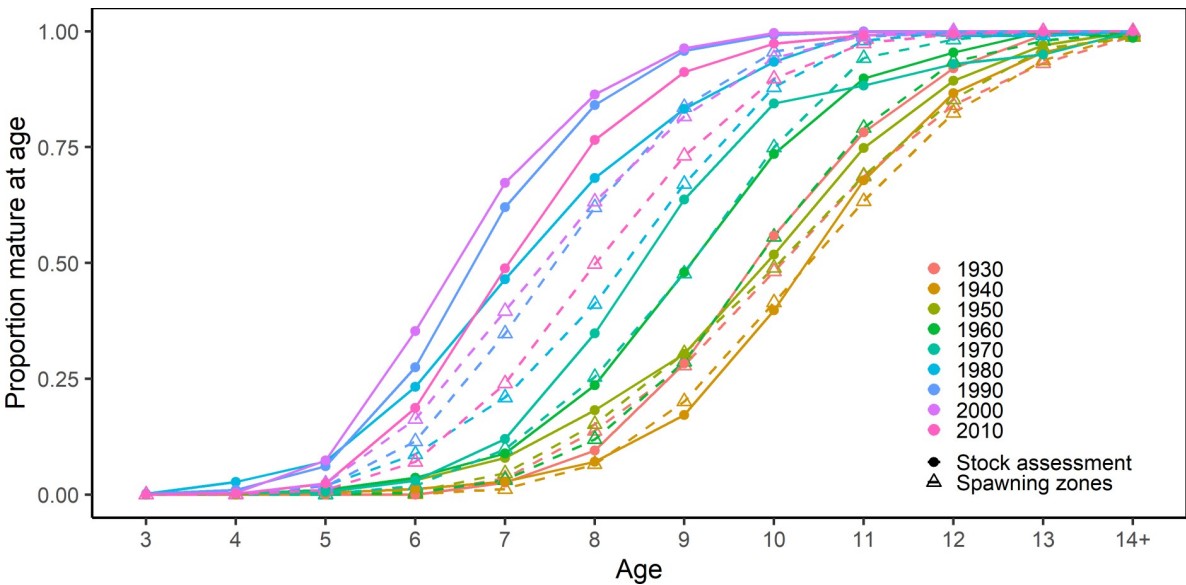

**Fig 6. Maturity ogives of NEA cod extracted from stock assessment and reconstructed using individual spawning zones, aggregated by decade.**

### Long-term analysis of maturity and spawning zone occurrence in NEA cod

The information extracted from the otoliths of a total of 382 218 fish collected on the spawning grounds during spawning season were used to reconstruct the maturity at age using spawning zones. The age composition of the fish in the archive changed between the late 1940s and the 2000s in parallel with changes in fishing pressure [35], with the proportion of old fish (age 12+) decreasing and the proportion of younger fish (mostly ages 6–8) increasing decade on decade (**S2 Fig**).

The maturity ogives from stock assessment showed a significant drift toward ealier maturation over time. In particular, there was a sudden large shift in maturity between the 1970s and the 1980s, as illustrated by the clustering of the subsequent decadal ogives (**Fig 6**). Reconstructing NEA maturity ogives based on individual increments showed similar temporal trends where a higher proportion of fish in all age classes were mature in later years, in particular since the 1960s and 1980s (**Figs 6 and 7**). Median age at maturity declined from 10.5 years in the period 1930–1960 to around 7.5 years in the 2000s (p<0.001, n = 85, Mann-Kendall trend test), with a subsequent reversal of this trend since 2010 (**Fig 8A**). When compared with stock assessment ogives, the reconstructed maturity time series was in close agreement in the period between 1930 and 1959, but showed similar trends with a temporal offset in the more recent decades (**Figs 6 and 7**). Although our reconstruction did not show the significant reduction in age at maturity in 1987, interquartile range did increase significantly, reflecting high individual variability in maturity for fish caught in that year (**Fig 8B**). Starting in the 1960s, the relationship between the proportion of mature fish calculated from spawning zones and the proportion derived from stock assessment showed the same convex shape as previously, where age classes at both extremes (either <10% or >90% of fish mature) had a very high agreement but the proportion of mature fish in the intermediate age classes was consistently lower using spawning zones (**Fig 9**). The reconstructed proportions of mature-at-age using individual spawning zones were consistently lower, however while the difference concerned mostly ages 8 to 10 during the 1960s it gradually moved to the ages 6 to 8 after 1982.

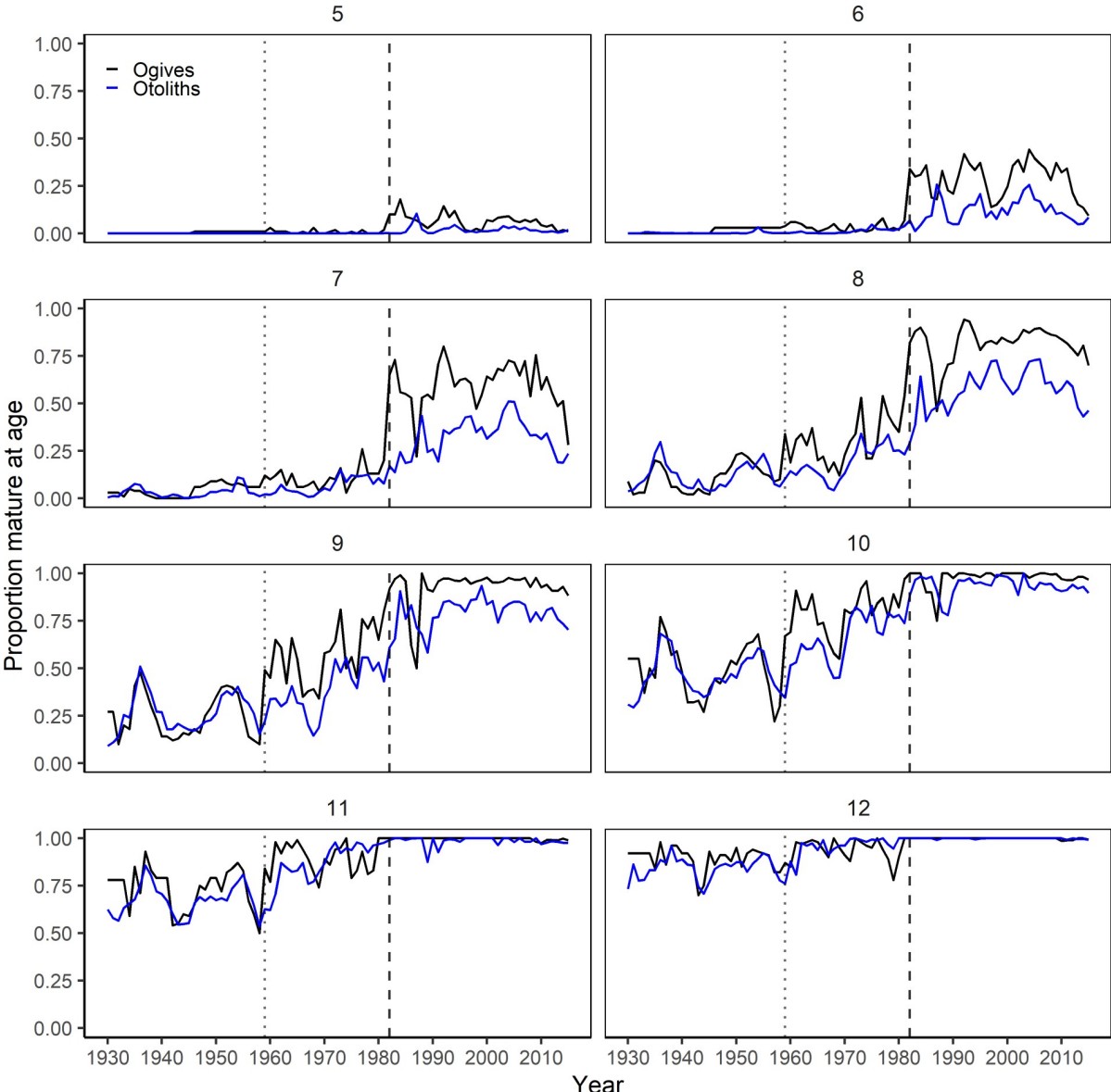

**Fig 7. Proportions of mature-at-age per year and age class from the official stock assessment ogives (black) and reconstructed for each year from individual increments identified as spawning zones (blue).** Vertical dotted line (1959) represents the beginning of the period combining Norwegian spawning zone data and Russian gonad readings for calculating maturity at age in stock assessment; vertical dashed line (1982) represents the beginning of yearly Barents Sea and Lofoten bottom trawl surveys and the change to using only gonad maturity stages in stock assessment.

Comparison of spawning zone interpretation between historical readers and a single contemporary reader for a subsample of age 8 fish showed evidence of changes in spawning zone interpretation over time. There was a high agreement between the mean age at maturity in subsampled fish and in the entire archive, indicating that our selection was representative of the larger collection and of the fish data used for assessments (**Fig 10**). The reinterpretation of spawning zones showed similar trends to the original reading, notably the significant decrease in age at maturity since the 1970s. However, mean age at maturity was consistently lower in the contemporary reading, with a difference of 0.2 to 0.5 years between 1950 and the 1990s

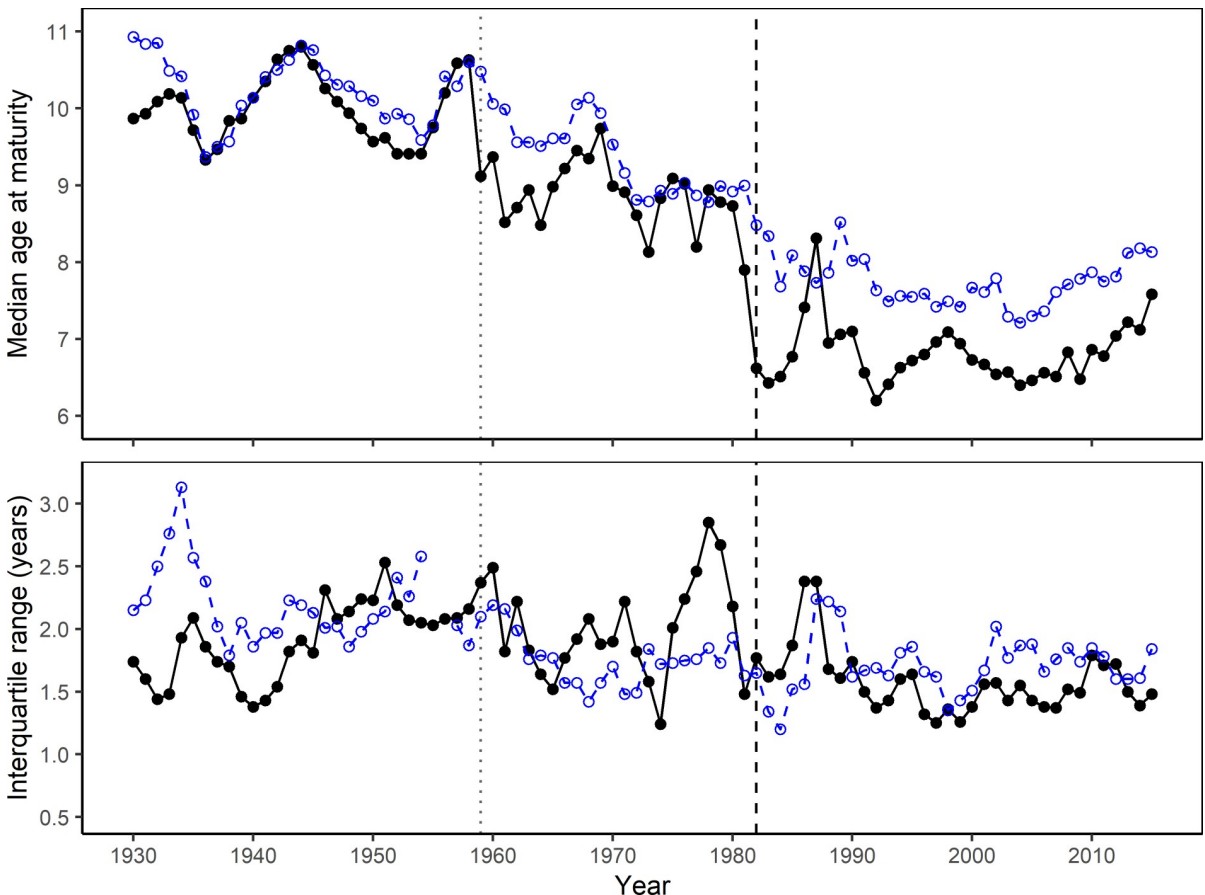

**Fig 8.** a) Median and b) interquartile range of age at maturity of NEA cod between 1930 and 2015. Calculated from the latest maturity ogives available from stock assessment (black solid line and points) and the proportions of mature-at-age reconstructed for each year from individual increments identified as spawning zones from the whole archive (blue dashed line and open points). See Fig 7 for vertical lines (1959 and 1982).

(Fig 10). This corresponded to a higher mean number of spawning zones identified by the contemporary reader with the biggest difference found between 1950 and 1990, while more recent years generally showed a high agreement (Fig 11).

## Discussion

### Occurrence of spawning zones in Atlantic cod populations

Rollefsen [14] first coined the term "spawning zone" to define specific otolith zonation patterns assumed to reflect individual spawning in Barents sea Atlantic cod populations almost a century ago. Despite their subsequent use in stock assessment and survey routines, these zones have since only seen anecdotal mention and interest outside of Norway (for example Greenland cod [42]), and their potential occurrence across different Atlantic cod populations has never been analysed systematically. Here, we provide evidence that spawning zones can be reliably identified in otoliths from multiple Atlantic cod populations characterized by a wide range of environmental conditions and life history modes. Previous experimental work had challenged the assumed relationship between spawning zones and individual reproduction in NEA cod due to the lack of consistent spawning-induced changes in the otolith structure [16, 17]. These studies have led to the suggestion that external factors such as stress and changes in

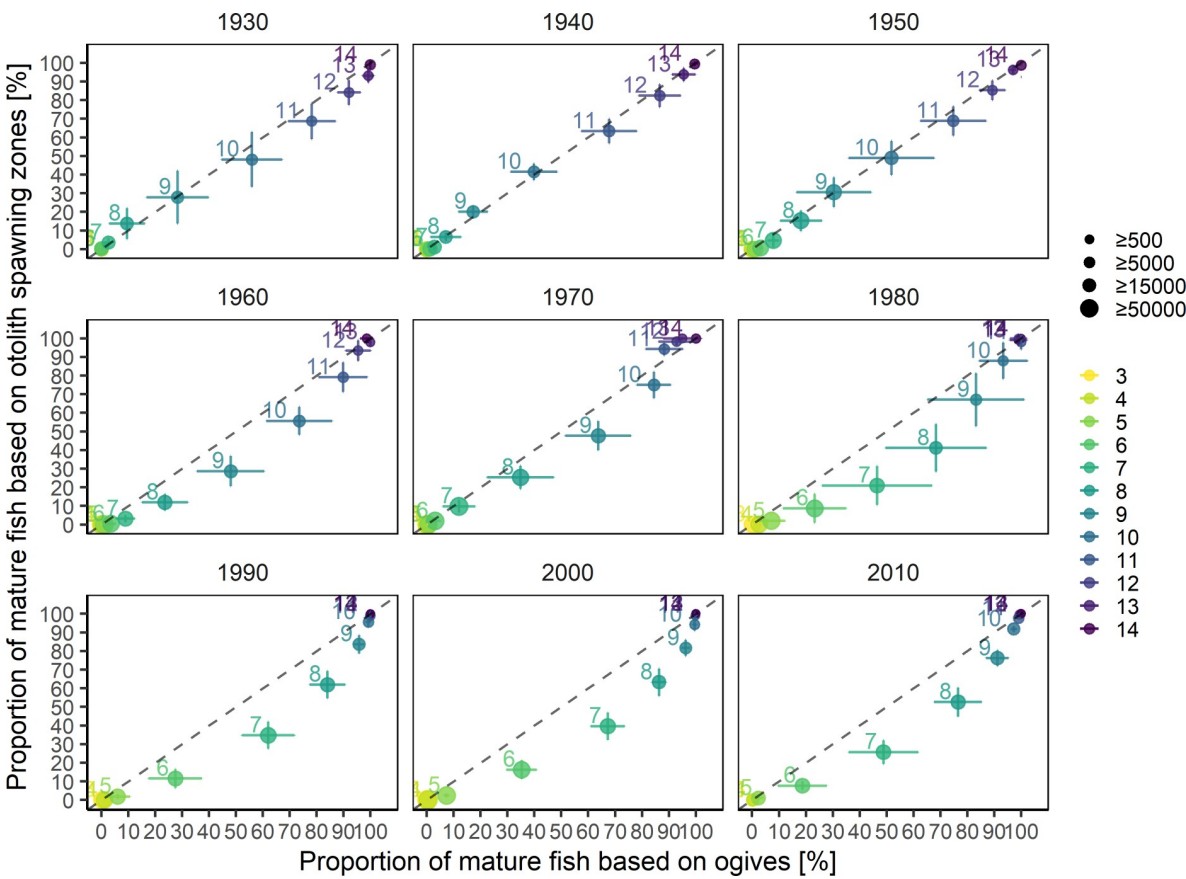

**Fig 9. Time series of maturity-at-age derived from spawning zones and from gonads.** Mean proportions of NEA cod mature-at-age (colors) calculated from individual increments from the whole archive identified as spawning zones for each year, compared to the mean proportions of mature-at-age calculated from the maturity ogives from stock assessment and aggregated by decade (facet). Error bars indicate the standard deviation of maturity ogives from spawning zones (vertical) and stock assessment (horizontal) within each decade. Point size is proportional to the number of individual increments per age class.

the environmental conditions associated with large-scale migrations could be key factors to inducing the formation of spawning zones. Our results show that even early maturing populations from highly seasonal areas such as the North Sea or Cape Breton commonly form spawning zones. In addition, the frequency and number of spawning zones increased with age and fit relatively well to the maturity curves constructed from gonad maturity stages, despite some age-specific population differences. Together, these findings suggest that spawning zones are a common trait to all Atlantic cod populations and that their occurrence is likely tied to changes in fish metabolism related to maturation, rather than a result of environmental influences stemming from migratory behaviour.

In Norway, similar spawning zones are also routinely read and identified in haddock (*Melanogrammus aeglefinus*) and saithe (*Pollachius virens*) otoliths [43], which suggests that spawning zones might be a broader trait of the family Gadidae. Given their importance in global fisheries [44], it would be interesting for further studies to examine whether other gadoids such as Alaskan pollock (*Gadus chalcogrammus*) or whiting (*Merlangius merlangus*) also form similar otolith checks, as they could provide new insights on maturity where historical otolith collections are available beyond the period traditionally covered by scientific surveys, and especially for data-limited stocks.

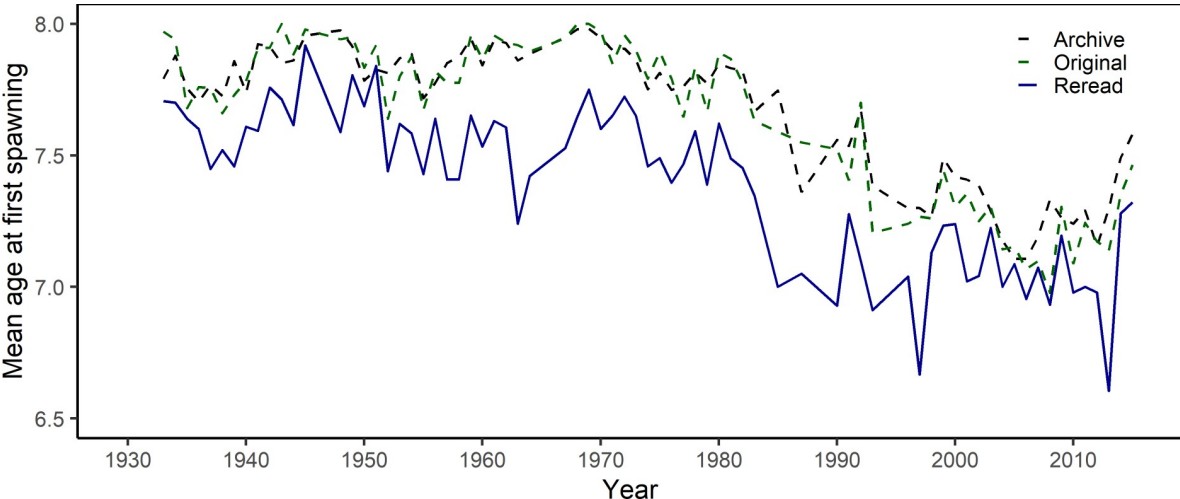

**Fig 10. Comparison of changes in age at maturity between readings.** Mean age at maturity of fish captured at age 8 calculated for each year between 1932 and 2015 from the entire archive (black dashed line), the original readings of our subsample (green dashed line) and the reinterpretation by a single contemporary reader (solid blue line).

## Relationship between spawning zones, maturity and reproduction

Both somatic growth and reproduction are complex biological processes directly dependant on the availability and allocation of surplus resources after basal metabolism and maintenance.

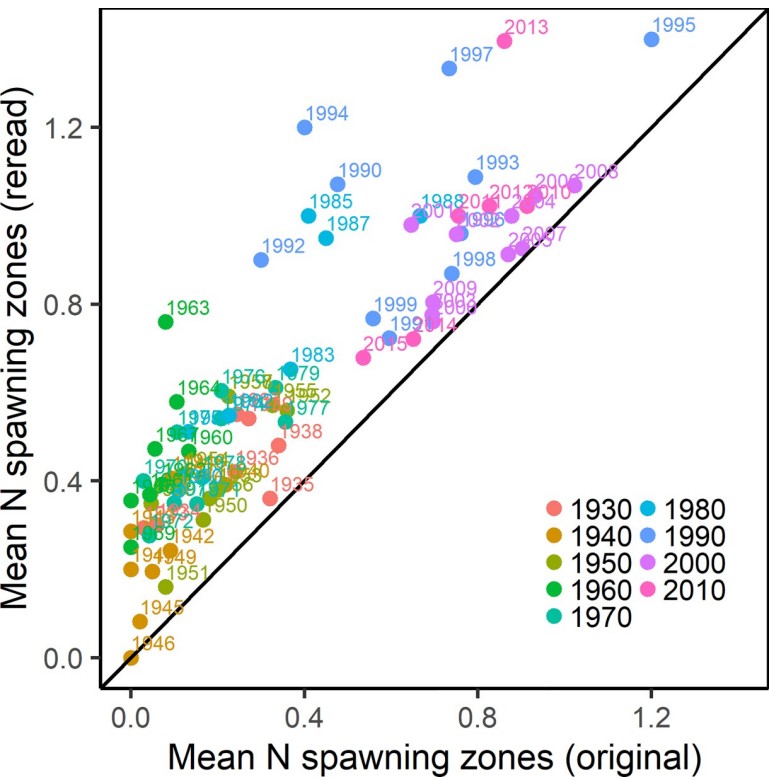

**Fig 11. Comparison of number of spawning zones between readers.** Yearly mean number of spawning zones in fish of age 8 calculated from the original and the reinterpreted subsample, decades coded by color.

Investment in one is generally at the expense of the other [45]. These trade-offs are evident in fish as they tend to follow indeterminate growth patterns, where increase in size is prioritized until the onset of sexual maturity, after which a significant proportion of resources is invested into reproduction and growth slows down [10, 46]. Since fish and otolith growth are generally correlated [47], this slower somatic growth following maturity likely affects otolith growth rates as well. Changes in energy allocation following maturity are also likely to affect the accretion rate and optical characteristics of otoliths due to changes in the metabolic activity and protein synthesis in particular [48–50]. Gonad development also induces metabolic changes which alter blood plasma and endolymph composition, in turn affecting otolith structure and chemical composition [51–53].

Significant changes in otolith growth patterns have been associated with the onset of sexual maturity in different fish species such as burbot (*Lota lota*), plaice (*Pleuronectes platessa*), orange roughy (*Hoplostethus atlanticus*) or sprat (*Sprattus sprattus*) [54–57]. However, while our results also indicate that the occurrence of spawning zones corresponds to maturity as assessed from gonads, we identified a singular discrepancy between the two for the intermediate age classes where only part of the population is mature. This disconnect was the largest for the earlier maturing populations, which suggests that the formation of spawning zones may differ between stocks with slow or fast life histories (or those living at the colder or warmer edges of the species range). This was further evidenced in our long-term reconstruction of NEA cod maturity ogives using individual increments as basis for calculating maturity at age. Indeed, the comparison with maturity assessed from gonad data for the period 1982–2015 showed similar interannual variations across all age classes but a consistent underestimation of proportion of mature fish at age for the age classes 6 to 9. The agreement was much better for the other age classes where the proportion of mature fish was either close to 0 (fish younger than 6) or close to 1 (fish older than 9). Similarly, the median age at maturity calculated from the reconstructed maturity ogives was consistently higher by about one year but the interquartile range remained similar, indicating that spawning zones correctly reconstructed the shape of the maturity curves but consistently underestimated the proportion of mature fish at age in comparison with gonad data.

The results of our analyses suggest that there may be a delay in the formation of spawning zones in the first years following maturity. Even under controlled experimental conditions, spawning zones are not formed consistently following reproduction [17]. Irgens *et al.* [17] suggested that spawning zones could form every year after first maturation, but may not necessarily represent spawning events. They also hypothesized that fish with low fecundity and reproductive investment such as first-time spawners may not experience significant enough metabolic changes to induce distinguishable otolith features. In Atlantic cod, first-time spawners are known to perform poorly and be characterized by a lower fecundity, a reduced egg production with smaller, lower quality eggs, and a relatively shorter breeding period [58]. It is therefore plausible that the lower success in spawning by recruit spawners, as well as the significant proportion of resources still utilized for somatic growth, does not affect their metabolism enough to change otolith growth and induce specific checks. Similarly, Folkvord *et al.* [16] commented on the apparent lack of spawning zones despite the visual confirmation of past spawning events based on post-ovulatory follicles. However, Folkvord *et al.* only used female fish of ages 6 and 7, where only 15 to 40% of the individuals are usually mature. Their observations could therefore be consistent with our hypothesis that the allocation of resources for reproductive investment during the years following first maturation may not be stable enough to induce spawning zones, as they only examined fish that had reached maturity a maximum of two years prior and therefore were unlikely to show any spawning zones outside of the marginal increment (which is incomplete and therefore harder to identify).

Another important aspect of cod reproductive biology is the high incidence of skipped spawning (up to 40%) seen in multiple cod populations [11, 22] and the absence of a matching interruption in otolith spawning zones. Skipped spawning in NEA cod is mostly characterized as a resting strategy where oocyte development is interrupted very early during the reproductive cycle, leading fish to skip their spawning migration and instead spend the spawning season on the feeding grounds [11]. Because the "decision" to skip spawning mostly seems to reflect low food availability and poor fish condition [25, 59], it was proposed that these factors influence the otolith growth similarly to reproduction and therefore lead to spawning zones appearing and being confounded with "normal" spawning zones even when fish skip spawning [17]. Alternatively, we propose that skipped spawning may not induce spawning zones altogether because the low energy levels are redirected toward fish maintenance and growth, and consequently do not lead to significant changes in otolith biomineralization. Since skipped spawning occurs predominantly among potential second-time spawners [24], it is reasonable to suggest that the spawning zone record really does reflect a 1–2 year offset between age at first maturity and number of spawning years. If first-time spawners generally have a poor reproductive investment and second time spawners may skip spawning as an investment in future reproduction, the spawning zones would be more likely to record actual successful reproduction events. In that sense, our results suggest that spawning zones may be more indicative of a profound change in fish metabolism and otolith deposition once fish condition is good enough to sustain repeated successful reproduction events (a "settling-down" of energy allocation), rather than a marker of "stress" associated with the energy costs of reproduction and migration.

## NEA cod spawning zones and changes in maturity throughout the last 100 years

The NEA cod, as well as other cod populations, has seen a significant decrease in age at maturity throughout the second half of the 20[th] century [60]. Our results show declining trends in maturity using spawning zones similar to those derived from gonad maturity stages. In the stock assessment, the Gulland method used for the period 1932–1981 included a correction for unequal mortality between mature and immature fish [38]. This was done in order to account for the suspected increased mortality once fish become mature, as spawning migrations are usually associated with higher stress and natural mortality as well as a higher vulnerability to fishing. In the present reconstruction we did not add any correction since we used the entire data at the individual level similarly to gonad-derived ogives, including also multiple individual contributions of older fish throughout their life. In relation to our previous points regarding first-time and skip spawning stress, the observed offset in maturity derived from spawning zones may indicate that first and second time spawners do suffer increased mortality and may therefore disappear from the population in the otolith archive. A reconstruction using individual spawning zones could therefore be biased by sampling and show a lower proportion of mature fish at age without any correction. This may contrast our previous discussion on the nature of spawning zone formation and the observed offset with maturity derived from gonads, but we suggest this is not necessarily the case here. Archived otoliths collected during surveys are taken from the same individuals whose gonads are read and used for stock assessment, and since the year 2000 additional samples have been provided by a "reference fleet" of selected commercial fishing vessels that collaborate with research institutes [61]. Unaccounted sampling bias due to the increased mortality of recruit spawners should be visible in both the gonad and spawning zone-derived maturity time series, which is not the case here. Therefore, the hypothesis of a biological delay between first maturation and spawning zone formation seems more likely to explain the observed differences.

The interpretation of spawning zones can be very subjective and variable between periods with different readers. For example, we show that a re-reading of historical material by a single contemporary reader consistently found more spawning zones, despite very similar maturity trends. This had previously been shown during otolith exchanges, where age reading of historical material was found to be consistent among contemporary readers, but they read more spawning zones on average compared to the historical readings [15, 62]. It is therefore plausible that some of the observed changes in spawning zone occurrence are representative of changes in spawning zone identification, and that our reconstructed maturity ogives for the earliest period may underestimate proportion of mature fish at age. While re-reading a near century's worth of archival material would require a considerable investment of time, we advise that, should spawning zones become more commonly used and studied, extra care be taken to ensure they are interpreted in a standardized way. Development of image analysis algorithms may help to improve the standardized collection of otolith visual information such as spawning zones.

Selective fishing is usually considered to be responsible for the significant decrease in age at maturity in cod populations, including NEA cod [41, 63, 64]. Fishing pressure has been shown to induce earlier maturation through two main mechanisms. First a "phenotypic response", as removing a portion of the stock reduces density-dependence and increases resource availability for the remaining individuals, which generally respond with faster growth and earlier maturation [41, 65]. Second, an "evolutionary response" where the sustained selective removal of late-maturing individuals before they reach maturity may favour early-maturation genotypes in the population [66]. The importance and respective contribution of each mechanism is however harder to disentangle, especially as they may express themselves on different time scales. NEA cod growth is largely density-dependant and fishing may therefore indirectly favour faster growth [27]. Thus, the present reconstructed maturity time series would support earlier maturation and faster growth as a response to density-dependence release. In the mid-1980s the population was critically depleted and a fishing moratorium was imposed [13]. New harvesting rules and management strategies were implemented that drastically reduced fishing mortality during the following decades [39]. However, despite a decreased exploitation pressure and a population biomass back to early 1950s levels, age at maturity has since remained relatively stable and has only shown signs of increasing since 2010. This suggests that the earlier maturation of NEA cod may reflect both a phenotypic response to density-dependence release and some fishing-induced evolution, as genetic changes potentially caused by fishing are not immediately reversed once earlier maturation has been selected in the population [66, 67]. Recent work has however shown no substantial genetic changes in the NEA cod based on the whole-genome sequence data obtained from the material collected before (early 20th century) and after (early 21st century) periods of intensive exploitation [68]. Warmer conditions have also been associated with earlier maturity in southern cod populations [69] and are likely to be one of the main factor behind the differences in age at maturity seen across the different cod populations [60]. It is therefore possible that high fishing pressure selected for earlier maturity throughout the period 1950s-1990s, but that the warming conditions in parallel may have continued and maintained the earlier maturation trend even after fishing was reduced. As shown by Marshall and McAdam [70], both scenarios are plausible for the NEA cod population but disentangling the respective influences on phenotypic plasticity in maturation is challenging when both drivers happens simultaneously and are likely interacting.

## Conclusion

In this study, we provide the first multi-population examination of the occurrence of spawning zones in Atlantic cod otoliths and their validity as proxies of individual maturity. Our results

show that spawning zones can be reliably identified in all Atlantic cod populations regardless of their environmental and life history characteristics, indicating that they are likely to be a general trait of Atlantic cod. Despite population-specific differences, the occurrence of spawning zones also seems to mirror maturity as determined from gonad examination. However, overestimated age at maturity derived from spawning zones for the intermediate age classes suggest a biological offset between gonad maturation and spawning zone formation, in particular for recruit and second time spawners. This suggests that profound metabolic and otolith deposition changes associated with an acclimated energy allocation are behind the formation of spawning zones. If true, then spawning zones could serve more widely as a powerful proxy for estimating individual reproductive contributions, even in other species.

## Supporting information

**S1 Fig. Maturity at age extracted from the current literature and stock assessment and aggregated by decade for each of the six cod populations.**
(TIF)

**S2 Fig. Proportions of archival NEA cod otoliths from each age class collected in the Barents Sea and at the Lofoten spawning grounds between 1932 and 2015, aggregated by decade.** Thick black line represents the average fishing mortality from ages 5 to 10.
(TIF)

**S1 Table. Summary statistics of reader precision for spawning zone interpretation.** Total of 41 samples were randomly selected across all 6 cod populations and read blindly in three individual instances by each reader. Median number of spawning zones found for each sample by each reader were then compared.
(DOCX)

**S1 Graphical abstract.**
(TIF)

## Acknowledgments

The authors thank Erlend Langhelle, Hildegunn Mjanger, Eirik Odland and Harald Senneset (IMR) for their insights on spawning zones and reading practices. We also thank Daniel Howell and Bjarte Bogstad (IMR) for the discussions regarding stock assessment of NEA cod. Finally, we thank Dr. Steven E. Campana, Dr. Peter Grønkjær and Dr. Ewan Hunter for providing cod otolith images for Canada, Greenland and the North Sea.

## Compliance with ethical standards

The authors declare they have complied with ethical standards.

## Author Contributions

**Conceptualization:** Côme Denechaud, Audrey J. Geffen, Szymon Smoliński.

**Formal analysis:** Côme Denechaud.

**Funding acquisition:** Audrey J. Geffen, Jane A. Godiksen.

**Investigation:** Côme Denechaud.

**Writing – original draft:** Côme Denechaud.

**Writing – review & editing:** Côme Denechaud, Audrey J. Geffen, Szymon Smoliński, Jane A. Godiksen.

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
