## [Decision Letter · Decision Letter 0]

16 Jul 2021

PONE-D-21-17385

Otoliths “spawning zones” across multiple Atlantic cod populations: do they accurately record maturity and spawning?

PLOS ONE

Dear Dr. Denechaud,

Thank you for submitting your manuscript to PLOS ONE. After careful consideration, we feel that it has merit but does not fully meet PLOS ONE’s publication criteria as it currently stands. Therefore, we invite you to submit a revised version of the manuscript that addresses the points raised during the review process.

Thank you for submitting a very interesting and timing manuscript. Please in your revision address the minor suggestions/comments made by reviewer #1.

We look forward to receiving your revised manuscript.

Kind regards,

Andrea Belgrano, Ph.D.

Academic Editor

PLOS ONE

Journal Requirements:

Reviewers' comments:

Reviewer's Responses to Questions

**Comments to the Author**

1. Is the manuscript technically sound, and do the data support the conclusions?

Reviewer #1: Yes

2. Has the statistical analysis been performed appropriately and rigorously? 

Reviewer #1: Yes

3. Have the authors made all data underlying the findings in their manuscript fully available?

Reviewer #1: Yes

4. Is the manuscript presented in an intelligible fashion and written in standard English?

Reviewer #1: Yes

5. Review Comments to the Author

Reviewer #1: This is an interesting manuscript that pulls together a huge amount of data to take a comprehensive look at a phenomenon referred to as spawning zones in Atlantic cod otoliths. Recent experimental work by others demonstrated that the formation of these marks does not necessarily correspond to the onset of spawning in cod. The present manuscript addresses this by comparing spawning marks from populations across most of the north Atlantic and testing them against direct measures of maturation status (gonadal examination). There are interesting trends decadally as well as regionally. One interesting conclusion is that spawning marks are a species trait, and rather than corresponding exactly to spawning, may rather reflect a "settling down" of the energy balance in a fish following maturity. Reasoned arguments are made in the Discussion about why these marks may occur, even when fish skip spawning to instead feed and regain energy.

One question I have is whether the authors can suggest why the translucent zones maintain their relative thicknesses, while the opaque zones become reduced. Yes, it is an energy balance, but what is it that causes no change in the translucent zones.

Minor edits:

- On line 225 the authors write "over 380 000 NEA cod" whereas everywhere else they write the exact number (382 218). Suggest they use the latter throughout.

- Line 243: insert "calendar" between "each" and "year"

- Lines 305-308, Figure 4 caption: please explain what the numbers on the columns refer to.

- Line 315 and elsewhere: "arcing shape" may not be the best descriptor. Consider using "convex" instead.

Overall, the authors are to be congratulated on such a comprehensive study.

6. PLOS authors have the option to publish the peer review history of their article (what does this mean?). If published, this will include your full peer review and any attached files.

Reviewer #1: **Yes: **Karin Limburg

---

## [Author Response · Author response to Decision Letter 0]

20 Aug 2021

We are very grateful for the positive reception and valuable comments provided in your review of our manuscript Otolith “spawning zones” across multiple Atlantic cod populations: do they accurately record maturity and spawning? 

We have carefully reviewed these suggestions and revised our manuscript accordingly. Our detailed answers to each point are provided in the document "Denechaud_response_to_reviewers" with the corresponding changes in the manuscript (“Denechaud_Revised_manuscript_track_changes.docx”).

---

## [Editor Report · Decision Letter 1]

26 Aug 2021

Otoliths “spawning zones” across multiple Atlantic cod populations: do they accurately record maturity and spawning?

PONE-D-21-17385R1

Dear Dr. Denechaud,

We’re pleased to inform you that your manuscript has been judged scientifically suitable for publication and will be formally accepted for publication once it meets all outstanding technical requirements.

Kind regards,

Andrea Belgrano, Ph.D.

Academic Editor

PLOS ONE

---

## [Editor Report · Acceptance letter]

3 Sep 2021

PONE-D-21-17385R1 

Otolith “spawning zones” across multiple Atlantic cod populations: do they accurately record maturity and spawning? 

Dear Dr. Denechaud:

I'm pleased to inform you that your manuscript has been deemed suitable for publication in PLOS ONE. Congratulations! Your manuscript is now with our production department. 

Kind regards, 

on behalf of

Dr. Andrea Belgrano 

Academic Editor

PLOS ONE